# Impact of Refinements to Handling and Restraint Methods in Mice

**DOI:** 10.3390/ani12172173

**Published:** 2022-08-24

**Authors:** Jennifer R. Davies, Dandri A. Purawijaya, Julia M. Bartlett, Emma S. J. Robinson

**Affiliations:** School of Physiology, Pharmacology & Neuroscience, Biomedical Sciences Building, University of Bristol, University Walk, Bristol BS8 1TD, UK; jd16203@bristol.ac.uk (J.R.D.); daryl.purawijaya@bristol.ac.uk (D.A.P.); jb16486@bristol.ac.uk (J.M.B.)

**Keywords:** mouse, stress, handling, restraint, refinement, cumulative suffering

## Abstract

**Simple Summary:**

For laboratory mice, handling methods used for routine husbandry and procedures can cause stress, particularly if tail handling methods are used. We first sought to investigate if there were any measurable impacts of different handling methods when tested in a controlled real-world study. Animals were handled at our breeding site using different methods, and their impacts on handling post-transport to the research site were analysed by animal care staff. We found a clear effect of handling method on technician scores relating to overt signs of stress and how easy they were to handle. Based on this study, the institution implemented a refined mouse handling policy. However, one barrier to the use of non-tail handling methods has been arguments that tail handling is necessary for restraint in mice. With a relatively minor adaptation to the conventional restraint method, we have been able to combine cupping with physical restraint for procedures avoiding any use of the tail. Here we provide data showing that our modified restraint method is associated with reduced signs of aversion in the mice. Together, our findings support the implementation of refined handling policies for mice, and this can include handling for both husbandry and restraint for procedures.

**Abstract:**

There is increasing evidence that, compared to non-aversive handling methods (i.e., tunnel and cupping), tail handling has a negative impact on mouse welfare. Despite this evidence, there are still research organisations that continue to use tail handling. Here, we investigated handling for routine husbandry by three different methods: tail, cupping and tube in a relevant real-world scenario involving mice bred off-site. After transfer to the destination unit, mice were assessed for overt behaviours associated with anxiety and fear. Mice that experienced tail handling were less easy to handle, were more responsive to the box opening, and scored lower in a hand approach test. One barrier to non-tail handling methods is the current practice of restraining mice by the tail for procedures. We therefore next assessed whether a modified method for restraint that takes the animal from cupping to restraint without the use of the tail was associated with better welfare. This refined restraint method reduced overt signs of distress although we did not find any differences in corticosterone levels or anxiety-related behaviours. These findings suggest that avoiding tail handling throughout the animal’s laboratory experience, including during restraint, benefits their welfare.

## 1. Introduction

Animals managed in a laboratory environment are usually handled on a regular basis for both routine husbandry and procedures. Physical restraint is still considered necessary for common procedures such as health checks, dosing and blood sampling, despite being shown to be aversive and causing negative affective state [1] as well as cardiovascular and hormonal changes [2,3,4]. Despite these regular handling interactions contributing to cumulative suffering, having a negative impact on the animals’ welfare and scientific outcomes, little relevance is given to such minor non-regulated stress related procedures.

Some efforts have been made to understand the source of handling stress and methods to minimise it. Familiarisation with handling and refined methods of picking up the animal leads to a substantial reduction in aversion towards the handler, the stress experienced during handling, and the anxiety this induces [5]. In particular, mice respond negatively to being picked up by the base of the tail causing aversion, higher anxiety levels, and they do not easily habituate to the method. Indeed, tail handling can cause seizures in susceptible strains [6]. In contrast to the traditional tail method, the alternative methods of cupping them in the hand or tunnel handling do not involve direct physical restraint of the animal. These methods make the animals easier to handle [7], increase voluntary interaction with the handler [5], and even after experiencing procedures such as dosing or oral gavage did not reduce their willingness to interact with a handler [8]. They also reduced stress and anxiety [5], induce reduced plasma corticosterone, reduce blood glucose with improved glucose tolerance [9] and improve responsiveness to sucrose reward [10]. These observations of a generalised reduction in stress and anxiety to non-aversive handling methods are robust, being seen across different institutions and laboratories [11,12,13].

Despite this evidence and the requirement under the 3Rs to use the most refined method, there are still research organisations that continue to use tail handling. One potential reason is that tail handling might be considered to be more efficient, and the perception that restraint methods require tail handling and so tail handling may be unavoidable. To investigate the impact of different handling methods, we used a relevant real-world scenario where mice bred off-site were handled for routine husbandry by either tail, cupping, or tube handling methods. Individual animal’s responses to handling following transport were then assessed by technical staff at the testing facility who had no prior knowledge of how the animals had previously been handled. This study and its design were requested by our AWERB in response to questions over the welfare benefits and practical challenges of implementing a no tail handling policy. We therefore focused the design on measures that were directly relevant to our technical staff. Although a potential limitation, we did not include sex or strain as factors in the analysis as these were not counter-balanced and the allocation of animals to handling method was made based on the breeding room they were located in. In the second part of the study, we investigated the impact of a modified restraint method that does not involve the use of the tail. We previously developed a refinement of the restraint method for intraperitoneal injections in rats [1] and apply a similar experiment design for this study. Mice were restrained using either a conventional tail handling method or our modified method that takes the animal from cupping to physical restraint without using their tail at any point in the procedure. Overt signs of stress and aversion were recorded as well as measures of affective state and the stress hormone, corticosterone (CORT).

## 2. Materials and Methods

### 2.1. Animals and Housing

For experiment 1, animals were mice bred at the University of Bristol breeding facility (GAA breeding stock and experimental animals which were a mix of wild type and specific mutant lines primarily on a C57BL/6 genetic background). The strain and sex were not counter-balanced or included as factors in the analysis. For the behavioural analysis of animals that were bred in an off-site facility a total of 212 animals were assessed. They were spilt into three different groups: tail handled n = 71, tube handled n = 42, cupping n = 99. As this experiment involved multiple research groups, details of group size, sex and age were not recorded and therefore not considered in analysis. For the assessment of the non-tail restraint method, subjects were male and female mice of two common laboratory strains: C57BL/6J and ICR (CD-1) (Envigo) and used animals available in the facility which were surplus stock to avoid the need to purchase animals specifically for this purpose. This does mean that strain, sex and age have not been fully considered in the analysis and is a potential limitation, but our primary objective was to determine the effects of handling at a species level. CD1 mice n = 40 and B6 mice n = 12 and for the CORT analysis ex breeder CD1 males n = 8 and B6 males n = 6 and females = 6. Voluntary interaction and elevated zero maze with subsequent plasma corticosterone analysis was performed with adult wildtype CD1 mice n = 16. A cohort of CD1 mice n = 12 were used for conditioned place preference. For the cotton bud test we used male CD1 n = 62 and B6 mice n = 32. For all behavioural experiments mice were randomly allocated to groups by odd or even identification number. Mice were housed at 12:12 h conventional light–dark cycle (lights on at 08:00 h) or reverse light–dark cycle (lights off at 08:00 h), in a temperature-controlled room (21 ± 1 °C) with relative humidity (45–65%). Animals were given standard housing conditions of cardboard tube, bedding material, plastic houses, and wooden blocks. Food and water were available *ad libitum*. All procedures were carried out under local institutional guidelines (approved by the University of Bristol Animal Welfare and Ethical Review Board) and in accordance with the UK Animals (Scientific Procedures) Act 1986. We also comply with ARRIVE guidelines. Details of the different animals that were used in both experiment 1 and the different tests which contributed to experiment 2 are detailed in Table 1.

### 2.2. Experiment 1: Technician Evaluation of Different Handling Methods 

To determine whether different handling methods result in objective changes in stress-related behaviour in mice, animals bred off-site were randomly allocated to three different handling groups defined by holding room. The facility consisted of multiple identical holding rooms and each room was allocated to a handling method to be used for all routine husbandry. Only offspring were included in the study to ensure that prior experience of different handling methods did not confound the results. (1) Tail—animals are picked up by the base of the tail and moved in the suspended position. (2) Cupping—animals were cupped using two hands to enclose the mouse to permit transfer. (3) Tube—animals were caught and moved using a tube (Figure 1). 

Mice were handled by these methods for all routine husbandry until transfer. Transport was undertaken in filtered transport boxes and animals travelled approximate 14 miles (30 min) by air-conditioned van. Number of mice per box for transport was between 1 and 6 animals and based on the individual researcher’s requirements for their studies. On arrival at the research facility and destination unit animals were unpacked by the receiving technician who scored the behaviour of the individual animals as detailed in Table 2. All mice were handled using the cupping method by the receiving technician who was blind to the animal prior handling method (colour coded identity cards). The scoring on arrival was undertaken by all staff in the receiving unit and all were trained in each method and followed a standard operating procedure agreed to ensure methods are applied in an equivalent manner.

Initial assessment was based on whole cage behaviour upon box opening. For the hand approach test, a gloved hand was lowered gently into the cage and rested on the substrate in the front half of the box without moving for 10 s; the willingness of the mice to approach the handler was then measured. For ease of handling individual animals were assessed. 

### 2.3. Experiment 2: Evaluation of a Modified Non-Tail Restraint Method 

This proposed refinement to the restraint method does not involve the use of the tail, which has been shown to induce aversion and high anxiety [5]. This method involves the mouse being removed from the cage floor either by cupping with one or two hands or by being picked up with a tube and tipped onto the hand. The mouse is placed onto the forearm and then completely covered gently with the other hand (Figure 2). When its head pokes out between the thumb and forefinger the animal is restrained by pinching the loose skin along the back of the animal between the thumb and forefingers in a similar way to a conventional tail handled restraint. 

For handling in experiment 2 all mice were cup handled for husbandry, and for experiments mice were cup handled onto the arm and then scruffed using the modified non-tail restraint into the experimental apparatus. Visual observations were recorded during and after release from non-tail and the conventional tail restraint method and as detailed in Table 3.

### 2.4. Voluntary Interaction Test

Here we assessed whether handling by the non-tail restraint method improves the willingness of mice to interact voluntarily with a handler compared to those restrained using the conventional tail method. This task was performed following the Hurst and West (2010) protocol. After removal of the cage lid and all environmental enrichment the handler stood unmoving directly in front of the cage for 60 s. After which a gloved hand was held resting on the substrate in the front half of the cage without moving for a further 60 s to assess interaction. Mice were then picked up using the cupping method and then either non-tail or tail restrained and placed back in their home cage. The handler then stepped back from the cage for 60 s and then repeated the two 60 s tests. Behaviour was recorded using a webcam and analysed offline using The Observer XT9 software [14]. Analysed behaviours were time spent on the front half of the cage and time spent interacting with the gloved hand (sniffing, climbing, biting, and putting limbs on the glove).

### 2.5. Elevated Zero Maze (EZM)

The EZM was used to assess anxiety-related behaviour. The EZM that was built in-house consisted of a grey, annular (55-cm diameter) runway (5 cm width) elevated (42 cm) above the floor. The runway was divided into 4 quadrants: 2 opposing ‘‘open’’ quadrants without walls (5-mm lip) and 2 opposing ‘‘closed’’ quadrants (20-cm high walls). Mice were placed in the closed quadrant of the maze, and activity was measured for 10 min and analysed offline manually, using either The Observer XT9 software [14] or BORIS v. 7.9.7 Software [15]. Time spent in the open quadrants, number of closed quadrant returns, and transitions between closed quadrants were recorded. The mice were defined as entering the other quadrant by the means of all four limbs entering the other arm.

### 2.6. Enzyme Immunoassay of Serum Corticosterone (CORT)

Blood samples for analysis of corticosterone were collected from two cohorts of mice exposed to tail versus non-tail restraint and the mice used in the EZM. Animals were killed by cervical dislocation 30 min following the restraint or the start of the EZM test. Trunk blood was collected in a 35 mm petri dish containing 50 µL 0.5 M EDTA (Invitrogen, product number 15575020) and 50 µL of Aprotinin (Sigma Aldrich, product number A6279). The sample was centrifuged for 10 min and corticosterone was measured in the plasma fraction by Radioimmunoassay [16]. Briefly, 20 µL of plasma was diluted in 480 µL of citrate-buffer (pH 3.0). Then, 100 µL of the solution was added with 50 µL of tracer solution (Oxford Bio Innovation DSL Ltd., Oxford, UK) and 50 µL of a specific rabbit anti-rat corticosterone primary antibody (kindly supplied by G. Makara, Institute of Experimental medicine, Hungary). Each sample was processed in triplicate. After overnight incubation in 4 °C, 500 µL of charcoal/dextran solution was added to each sample. Samples were then centrifuged (4000 r.p.m. for 15 min at 4 °C, 3.120 g) and aspirated before measured using a gamma counter.

### 2.7. Conditioned Place Preference (CPP)

Conditioning was conducted in a three-compartment apparatus (CPP box 75 cm wide × 30 cm deep × 20 cm high) made of clear Perspex which was built in-house. The middle (neutral) compartment (13 × 30 × 20 cm) had two manual guillotine-style doors each leading to a conditioning compartment (30 × 30 × 20 cm). One conditioning chamber had chopped timothy hay flooring the other had woodchip. **Conditioning:** The mice received 8 days of conditioning training. Tail handling was associated with one type of floor substrate, cupping was associated with the other floor substrate. During the conditioning training, the mouse was picked up for 10 s and then released back into the CPP for 90 s. This was repeated 5 times then the mouse was returned to its home cage. The arena was rinsed with 70% ethanol and dried between animals. **Test day:** Mice were placed in the neutral compartment of the arena with both doors open and allowed to move freely for 30 min. Animals were recorded and left undisturbed during the testing period.

### 2.8. Cotton Bud Biting Test

The cotton bud biting test was used as a measure of non-social aggressive behaviour [17]. The mouse was restrained and held facing the experimenter. A cotton bud was then presented in front of the mouse’s mouth. The cotton bud was presented 10 times with 5 s gaps between presentations. Total amount of trials where biting occurred and duration of each bite with a maximum of 5 s were recorded. New cotton bud was used for each animal.

### 2.9. Data Analysis

All data were analysed using SPSS 24 (SPSS, Armonk, NY, USA). Evaluation of different handling methods and refinement of restraint method were analysed using a Kruskal–Wallis test with Post Hoc Mann–Whitney U test. Plasma CORT, EZM and cotton bud test were analysed using an unpaired *t*-test. Voluntary interaction test data were analysed using a Repeated Measures ANOVA with WEEKS as within-subject factor, and GROUPS and SEX as between-subject factors. CPP was analysed using a one-sample *t*-test against 0 for column analysis and a paired *t*-test for habituation vs. testing day preference analysis. All significance tests were performed at alpha level of 0.05 and where significant interactions were identified in the main ANOVA a post hoc analysis was performed.

## 3. Results

### 3.1. Experiment 1: Effect of Off-Site Breeding Facility Handling Methods on Overt Behaviours at Destination Unit

Here we investigated behaviour in mice in a real-world relevant scenario, where we analyse the effect that different handling methods used at the breeding facility have on overt behaviours scored by animal care staff during receipt at the destination unit (full data Appendix A). Mice handled by cupping and tube methods show higher scores for aversive response to box opening (Main effect of handling method Kruskal–Wallis, *p* < 0.0006, Figure 3A). Mice handled by cupping and tube methods are easier to handle during unpacking than mice handled by the tail (Main effect of handling method, Kruskal–Wallis, *p* < 0.0001, Figure 3C). Mice handled by cupping were less likely to approach during the hand approach test when compared with tube or tail handling (Main effect of handling method Kruskal–Wallis, *p* < 0.0008 Figure 3E). Post-hoc pairwise comparisons showed that mice handled by the tail showed a significantly higher aversive response than those handled by cupping or tube in the ease of handling and response to opening the box. However, but there was no difference between scores from mice handled using either the tube or cupping method.

### 3.2. Experiment 2: Modified Restraint Method Reduces Overt Behaviours Associated with Aversion and Stress

Behaviour observations revealed a main effect for struggling (Kruskal–Wallis, *p* < 0.0001, Figure 4A), vocalisation (Kruskal–Wallis, *p* = 0.0121, Figure 4B) and aversion to release (Kruskal–Wallis, *p* < 0.0001, Figure 4E). There was no significant difference in faecal count (Kruskal–Wallis, *p* = 0.6614) or urination (Kruskal–Wallis, *p* = 0.0742). Post-hoc pairwise comparisons also revealed for struggling there was a significant difference between restraint method for mature CD1 males (*p* = 0.0051). For aversion to release there was a significant difference in restraint method for the CD1 males (*p* = 0.0286), CD1 females (*p* = 0.0286) and mature CD1 males (*p* = 0.0101).

As the individual group numbers were relatively small and each subgroup was fully counter-balanced, results for all animals were pooled for each method. The resulting grouped data showed that the refined restraint method was associated with a reduction in struggling (Mann–Whitney, *p* < 0.0001, n = 22 per group, Figure 4A), vocalization scores (Mann–Whitney, *p* = 0.0186, n = 22 per group, Figure 4B) and aversion to release (Mann–Whitney, *p* < 0.0001, n = 22 per group, Figure 4E). Blood samples collected from animals restrained using the two different methods did not show any differences in plasma CORT Figure 4F.

### 3.3. Tail versus Non-Tail Restraint Had No Effect on Voluntary Interaction

To assess behaviour in anticipation of handling, we assessed voluntary approach and interaction with the handler before and after handling on specified test days. Assessment both immediately before and after handling in each session allowed us to examine the change in behaviour immediately after handling as well as longer-term changes between sessions as animals became more familiar with handling. Interaction with the handler significantly increased with day Figure 5A (F_(2,24)_ = 8.609, *p* = 0.002) as they become more familiar with the experience without significant differences between restraint methods (F_(1,12)_ = 1.593, *p* = 0.231) or between male and female (F_(1,12)_ = 3.523, *p* = 0.085). There were no interactions between day and different restraint methods (F_(2,24)_ = 1.476, *p* = 0.249) and between day and sex (F_(2,24)_ = 2.887, *p* = 0.075) on the voluntary interaction behaviour. Approach on the observer-side of the cage showed no significant effect of day (F_(2,24)_ = 2.483, *p* = 0.105) and sex (F_(1,12)_ = 2.618, *p* = 0.132) but with the tail-restrained mice approaching the observer-side of the cage more than the non-tail restrained group (F_(1,12)_ = 6.170, *p* = 0.029). There were no interactions between day and different restraint methods (F_(2,24)_ = 1.333, *p* = 0.282) and between day and sex (F_(2,24)_ = 2.615, *p* = 0.094) on the approach behaviour.

All mice showed the expected pattern of behaviour, spending less time on the anxiogenic open quadrants compared to the closed quadrants. However, different restraint methods did not affect the time spent in the open quadrant of the zero maze Figure 5B (t (14) = −0.386, *p* = 0.705). We similarly found a lack of effect on latency to enter the open quadrants Figure 5C (U = 28.00, *p* = 0.7209) and the number of transitions between enclosed quadrants Figure 5D (U = 22.50, *p* = 0.3276). At the end of the experiment, restraint method did not significantly alter plasma CORT Figure 5E (U = 29.00, *p* = 0.776) after exposure to EZM.

### 3.4. Different Restraint Methods Have No Effect on Behavioural Measures of Anxiety or Aggression

CPP test showed a significant change in preference to the right hand-side chamber from habituation phase to the testing phase Figure 6A (t_(11)_ = 3.048, *p* = 0.011). However, during the testing phase, there were no significant preferences to either substrates (one sample *t*-test t_(11)_ = 0.044, *p* = 0.965), chamber sides (one sample *t*-test t_(11)_ = 1.020, *p* = 0.329), or restraint method (one sample *t*-test t_(11)_ = 1.774, *p* = 0.103).

There were no differences in the number of bites between mice handled by tail or non-tail restraint methods for the B6 or CD1 mice Figure 6B. However, the biting frequency in the B6 was higher than the frequency observed in the CD1 animals.

## 4. Discussion

Our present study demonstrates that non-tail handling methods can benefit mouse welfare and ease of handling when methods were compared in a real-world scenario. The method of handling during pre-study husbandry at an off-site facility affects the animals’ subsequent response to being handled. We showed significant effects on overt behaviours associated with stress and anxiety on arrival where tail handled mice were less easy to handle during unpacking and were more responsive to the box opening than cup or tube handled mice. The only measure that did not indicate a lower level of aversion was the hand approach test, where the cupping handled mice scored lower in the hand approach test compared with tube or tail handled mice. Our results confirm previous laboratory-based studies in selected study populations and show that, in a real-world setting, cupping or tube handling are potentially beneficial for both the mouse and the animal care staff. The outcomes of this study were reported to the local AWERB and formed the evidence base to support the implementation of a no tail handling policy for routine husbandry. The availability of locally obtained evidence also benefit the engagement of all staff in supporting the new policy. We also describe an alternative restraint method that can be combined with non-tail handling methods enabling implementation of a no tail handling policy across both husbandry and procedures. Our modified restraint method did not have any negative impacts for the animal and overt signs of distress were reduced in animals experiencing non-tail restraint. Similar outcomes were seen when animals that experienced different handling methods were testing in tasks relating to affective state-related measures and quantification of plasma corticosterone did not find any differences between methods [13].

Tail handling is considered aversive to rodents and has been hypothesised to resemble the feeling of being caught by a predator [7]. An alternative idea relates to conspecific fight bouts that target the dorsal and ventral areas around the tail [18], and hence mice may react to human handling by the tail as a form of aggression. Whether linked to predation or aggression, tail handling is likely to trigger an instinctive reaction to defend/avoid these interactions and generate a stress response. Although studies have previously reported the impacts of different handling methods on welfare measures, we wanted to determine if these non-aversive handling methods would generate a behavioural difference that could be quantified by animal care staff handling the animals. This type of experiment potentially has more relevance and can be used to provide local, quantitative evidence to support the implementation of mouse handling policies. Mice are bred at an off-site facility, and for the purpose of the study were handled by different methods from weaning until they were transported to the destination unit where they were behaviourally assessed. The tail handled mice were more responsive to box opening. All mice were then cupping handled and scored on the basis of their response to this method. Tail handled mice were less easy to handle. Whilst some animals were previously cupping handled, which may impact on the response to the scores for response to handling, we also saw changes in the measures taken before handling, i.e., response to box opening would suggest a reduced stress response in the animals that were not tail handled. We did not find the same effects in the hand approach test with cupping mice showing lower scores, although this may relate to the way the test was carried out relative to previous publications, or transport stress immediately preceding the assessment may impact their baseline stress levels and mask more subtle behavioural outputs [7].

We have shown that handling methods at off-site facilities can have measurable effects on technician scores of anxieties and fear-related responses of the animals. Perhaps most important though was that these data show that non-tail handling methods make the mice easier to handle, and so could overall benefit the efficiency of the management of the animals during routine husbandry. Easier to handle mice and reduced distress associated with human contact may also be beneficial during more stressful procedures such as tail bleeds and gavage are less distressed and therefore potentially less consequences on hormonal disturbances. Implementation of non-aversive methods may be considered less feasible if they are time consuming especially when dealing with potentially hundreds of rodents in a breeding facility. However, these data suggest that the investment of time in using non-tail handling methods during an animal’s early life can benefit their long-term behaviour towards the handler. This is also consistent with recent data that has shown that even brief fortnightly familiarisation with non-aversive handling methods during cage cleaning is enough to show increased exploratory behaviour and robust habituation/dishabituation which is shown to be consistent with decreased anxiety [19,20]. Therefore, implementation of non-aversive handling during routine husbandry would be an effective and simple way to reduce handling distress with no extra time costs and may even save time overall. There are a number of limitations with this study design including the lack of data relating to sex, strain or age difference. However, Hurst et al. have shown that non-aversive handling methods are beneficial on discrete measures. This experiment was designed to be a simple real-world study to support the assessment at an institutional level of the benefits of different handling methods to implement policy change. At a population level, this study does suggest benefits to animal welfare.

In our restraint study, animals showed lower levels of overt behaviours including struggling, vocalization and aversion on release when using our modified technique. As strains and sexes can differ substantially in anxiety and stress responsiveness [21,22] we assessed responses of both sexes for the inbred mouse strain C57BL/6, and the outbred strain ICR(CD-1). Female and male mice showed similar responses to the different restraint methods. Further, both strains showed the same general pattern of behaviour supporting our proposal that non-tail restraint benefits mice irrespective of strain or sex. However, in the EZM that measures anxiety we observed no behavioural differences in all parameters measured between the tail and non-tail restraint methods. This was consistent with Gouveia and Hurst et al. 2019 [7] where no differences in the time spent in open arm was observed. However, Gouveia and Hurst et al. 2019 [7] did observe that mice picked up by the tail showed a higher frequency of protected stretch attend postures compared to both tunnel and cup handled mice. We did not record these behaviours and so are unable to make a direct comparison. This observation by Gouveia and Hurst et al. 2019 [7] suggests that anxiety induced in the EZM does not mask the more subtle effects of the different handling methods. We also measured plasma CORT 30 min after the EZM as a measure of stress response. There was no significant effect of handling method on plasma CORT taken after EZM. Measures of CORT after different handling methods have mixed results with studies finding no significant effect of cupping versus tail handling method on faecal corticosterone metabolites [13]; conversely, lower stress induced plasma corticosterone concentrations have been observed in cupping handled mice versus tail handled [9]. These differences may be due to methodology, and also increases in plasma glucocorticoid concentration in response to stress is superimposed on the circadian secretory pattern making accurate measures of plasma CORT in response to mild acute stress difficult. Previous studies have also suggested that CORT may not be the most reliable method to quantify distress in mice [23] and previous studies have similarly reported a dissociation between CORT levels and other measures of distress [24,25]. In the cotton bud biting test there was no significant difference in the biting frequency between tail and non-tail restrained mice, suggesting that handling method had no effect on levels of aggression related behaviour. However, the CD1 mice showed a lower biting frequency compared to the B6 mice which contrast with inter-species aggression where CD1 mice have been shown to be more aggressive [26,27]. This suggests that there are strain differences in this particular test, but the reasons for this could not be further interpreted from this data set.

Whilst tail handling is currently the most widespread method for restraining mice, these findings show that restraint can be achieved in a way that is compatible with non-tail handling policies. Not only does this modified method mean that tail handling can be avoided throughout the lifetime experience of the mouse; we also show that it generates less overt signs of aversion and distress and is therefore also more refined than conventional methods. These studies utilised animals that were surplus from other studies and therefore did not fully address questions about sex or strain difference; further studies using fully counter-balanced designs are needed to better understand how handling methods are influenced by these factors. However, as with experiment 1, these studies suggest that a non-tail handling method of restraint may reduce distress and benefit both animal welfare and scientific objectives.

## 5. Conclusions

Exploring ways in which the cumulative lifetime experiences of an animal can be improved is both a legal and ethical requirement for animal research. The long-lasting negative effects of tail handling techniques has been consistently observed in terms of behavioural and physiological responses in carefully controlled studies. Generating local data on the impacts of different handling methods has provided a robust framework for implementing a policy of no tail handling for all routine husbandry at our institution with support from our local AWERB. Similar study designs could readily be used by other organisations without the need for specialist skills or equipment and generate data to support similar policy change. We also demonstrate here that a refined restraint method which does not involve use of the tail can be used alongside such a policy to further refine animal experience during procedures involving restraint. By reducing this source of distress, we can potentially benefit mouse welfare however, longer-term studies of affective state effects are needed [1,28]. There is also the potential to improve reproducibility and variability of scientific data and thereby reduce animal numbers.

## Figures and Tables

**Figure 1 animals-12-02173-f001:**
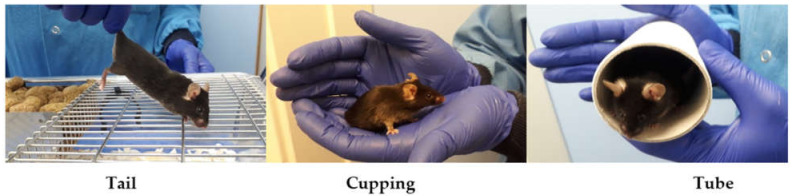
Demonstration of different handling methods.

**Figure 2 animals-12-02173-f002:**
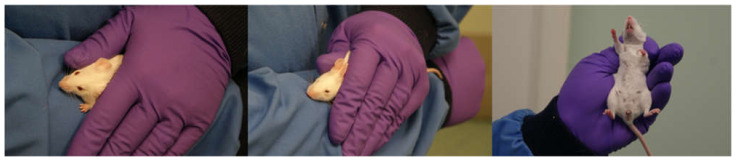
Demonstration of handling technique for non-tail restraint of mouse.

**Figure 3 animals-12-02173-f003:**
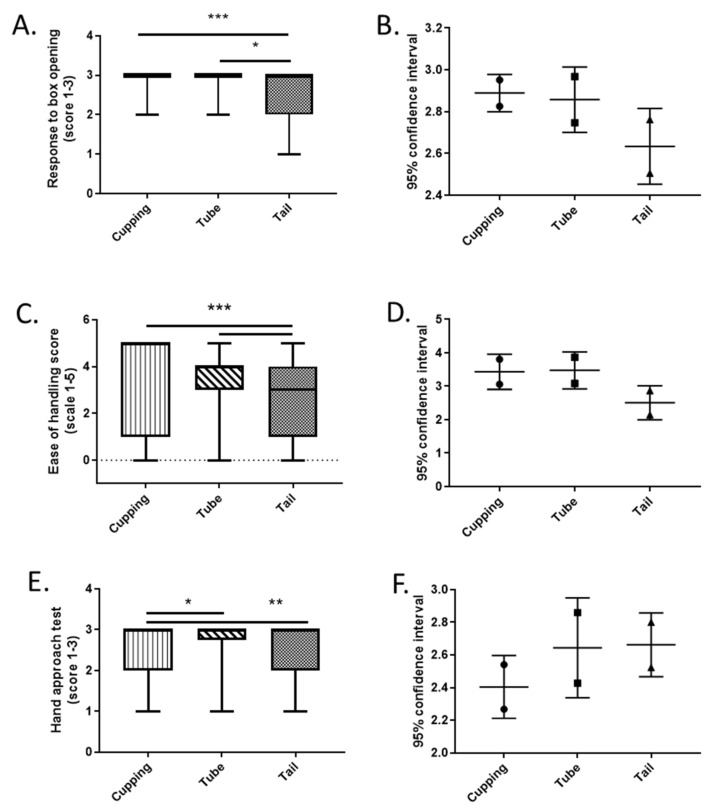
Effect of off-site breeding facility handling methods on overt behaviours at destination unit. (**A**,**B**) Mice handled by cupping and tube methods show lower scores for aversive response to box opening than mice handled using the tail. There was a main effect of handling method (Kruskal–Wallis ANOVA *p* < 0.0006) 3 = calm, minimal response, 2 = some agitation, 1 = obvious escape attempts. (**C**,**D**) Mice handled by cupping and tube methods are easier to handle during unpacking than mice handled by the tail. There was a main effect of handling method (Kruskal–Wallis ANOVA *p* < 0.0001) Ease of handling: 5 = easy, 4 = mild escape attempt, 3 = moderate escape attempt, 2 = escaped, 1 = jumped from hand, 0 = not possible to cup. (**E**,**F**) Mice handled by cupping were less likely to approach during the hand approach test when compared with tube or tail handling. There was a main effect of handling (Kruskal–Wallis ANOVA *p* < 0.0008). 3 = one or more animals approached hand, 2 = no obvious response, 1 = active avoidance. For all groups n = 97 tail handled, n = 42 tube handled, n = 71 cupped handled. Data shown as mean ± S.E.M (left panel) and 95% confidence interval (right panel). (* *p* < 0.05, ** *p* < 0.01, *** *p* < 0.001).

**Figure 4 animals-12-02173-f004:**
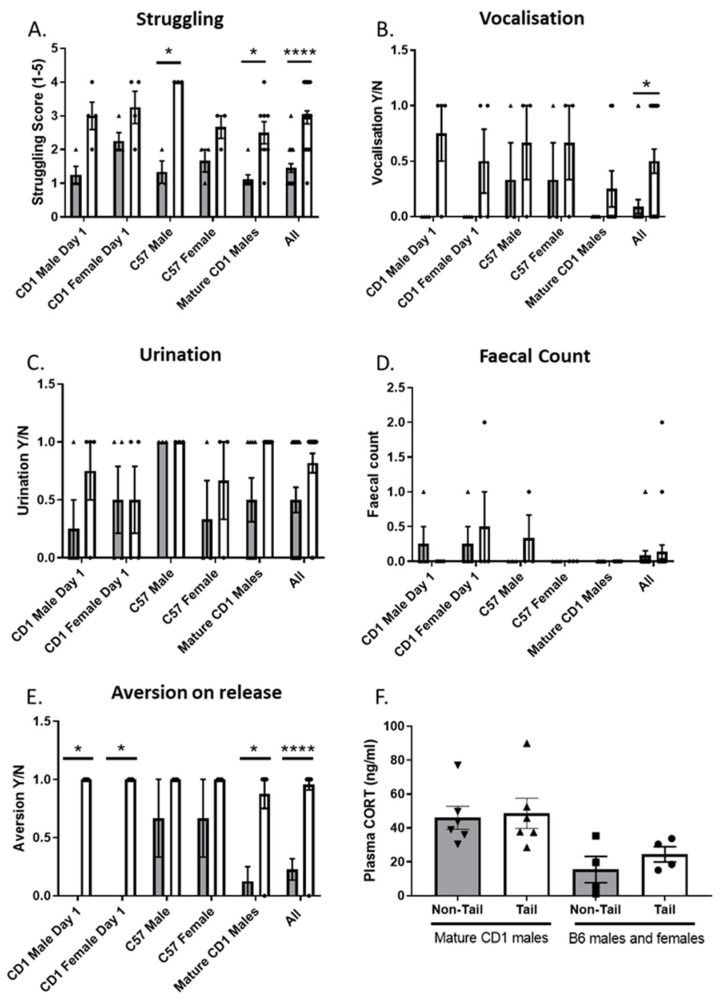
Effect of handling method on overt behaviours following different restraint methods. (**A**) Mice handled by non-tail restraint showed lower scores for struggling (Kruskal–Wallis ANOVA *p* < 0.0001), (**B**) Mice handled by non-tail restraint showed lower occurrence of vocalization (Kruskal–Wallis ANOVA *p* < 0.0121), (**C**) lower occurrence of urination, (**D**) lower average faecal count, (**E**) lower scores for aversion to release (Kruskal–Wallis ANOVA *p* < 0.0001). For groups CD1 males *n* = 16 (tail = 8, arm = 8), CD1 females n = 16 (tail = 8, arm = 8), B6 males *n* = 6 (tail *n* = 2, arm *n* = 4), B6 females *n* = 6 (tail = 4, arm *n* = 2), CD1 mature males *n* = 8 (tail = 4, arm = 4). (**F**) No effects on CORT were observed CD1 males (tail = 4, arm = 4) and B6 male and female (tail *n* = 6, arm *n* = 6). Data shown as mean ± S.E.M. (* *p* < 0.05, **** *p* < 0.0001). Grey bar = non-tail restraint, White bar = tail restraint.

**Figure 5 animals-12-02173-f005:**
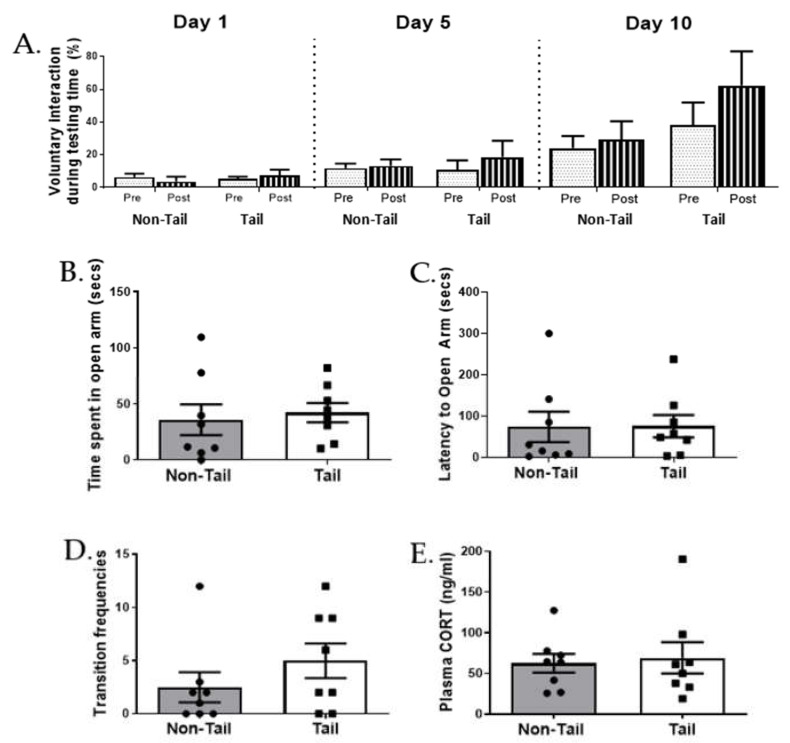
Restraint method has no effect on voluntary interaction with handler or elevated zero maze. Voluntary interaction with the handler before and after different restraint methods. (**A**) Percentage of test time interacting with the handler immediately before (pre, dotted bars) and after (post, stripped bars) the first, fifth and tenth handling session for CD1 mice restrained by tail or non-tail methods. Behaviours in the elevated zero maze after different restraint methods. (**B**) time spent in open arm, (**C**) latency to open arm, (**D**) transition frequencies, (**E**) plasma corticosterone. For all groups n = 16 CD1 males n = 8 (tail = 4, non-tail = 4) and females n = 8 (tail = 4, non-tail = 4). Data shown as mean ± S.E.M.

**Figure 6 animals-12-02173-f006:**
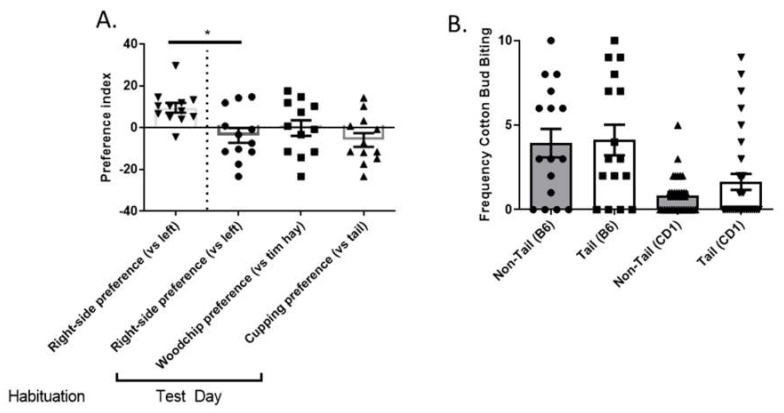
Different restraint methods have no effect on behavioural measures of anxiety and aggression. (**A**) Mice preferences to handling methods on conditioned place preference test. (**B**) biting frequency in cotton bud task. A cohort of CD1 (male = 12) were used for conditioned place preference. For the cotton bud test we used male CD1 n = 62 (tail n = 31, non-tail n = 31) and C57BL/6J mice (male n = 16 (tail n = 8, non-tail n = 8) and female n = 16 (tail n = 8, non-tail n = 8)). Data shown as mean ± S.E.M. (* *p* < 0.05).

**Table 1 animals-12-02173-t001:** Experimental Timeline.

Cohort	Test (in Order)	Strain	Age	Male	Female	Housing
		CD1	8 months	16	16	group (2)
	Modified Restraint Method	mature CD1	11 months	8	0	singly
A		B6	8 months	6	6	singly
	Modified Restraint CORT	B6	8 months	6	6	singly
		mature CD1	11 months	8	0	singly
subset A	Voluntary interaction test	CD1	8 months	8	8	group (2)
	Elevated zero maze					
	Elevated zero maze CORT					
B	Conditioned place preference	mature CD1	11 months	12	0	singly
C	Cotton bud test	CD1	mixed age	62	0	singly/group
		B6	mixed age	16	16	

**Table 2 animals-12-02173-t002:** Visual overt behavioural observations during initial assessment upon arrival at research facility.

Initial Assessment Upon Box Opening	Hand Approach Test	Ease of Handling
3 = calm, minimal response 2 = some agitation 1 = obvious escape attempts	3 = one or more animals approach hand 2 = no obvious response 1 = active avoidance	5 = easy 4 = mild escape attempts 3 = moderate escape attempts 2 = escaped1 = jumped from hand 0 = not possible to cup

**Table 3 animals-12-02173-t003:** Visual overt behavioural observations during non-tail and the conventional tail restraint methods.

Visual Observations	
Struggling effort to be released from grip	
(1) No struggling once restrained	
(2) Slight struggling for a short period of time	
(3) Slight struggling throughout/moderate struggling for a short period	
(4) Moderate struggling throughout/severe struggling for a short period	
(5) Severe struggling throughout	

Vocalisations made during grip	Yes = 1, No = 0
Urination during or after grip	Yes = 1, No = 0
Escape behaviour (indicated by running or avoiding hands)	Yes = 1, No = 0

## Data Availability

Data is available on request from the corresponding author.

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
