# Peer review of "Impact of Refinements to Handling and Restraint Methods in Mice"

_animals, 2022, doi:10.3390/ani12172173_

Round 1

Reviewer 1 Report (Previous Reviewer 3)

Lines 23,53, 482 change tunnel to tube, and elsewhere. Maybe explain somewhere that tube is the same as tunnel used on other publications.

Line 409: affects not effects 

Line 451. The effect of handling, of omitted

Line 457: are less distressed, not ' ...causing less distressed...'

Author Response

Thankyou for the opportunity to respond to the reviewer’s comments on our manuscript. We also appreciate the time the reviewers have taken to provide helpful feedback and suggestions of how we can improve the presentation of the work. We have provided a response to each of the points raised below and also made changes to the manuscript which are highlighted in the revised version. We hope that these will address the reviewer’s concerns and the paper is now suitable for publication.

Lines 23,53, 482 change tunnel to tube, and elsewhere. Maybe explain somewhere that tube is the same as tunnel used on other publications.

All references of the tunnel have now been changed to tube for consistency. We also have a picture of the tube in figure 1 for clarity

Line 409: affects not effects 

Line 451. The effect of handling, of omitted

Line 457: are less distressed, not ' ...causing less distressed...'

We have also changed the mistakes highlighted in the paper.

Reviewer 2 Report (Previous Reviewer 2)

This is a new/re-submission of submission 1372745. Some of the comments and suggestions made have been addressed but there are still several short comings some major other minor. 

M&M:

Although the authors state that they 'comply with ARRIVE guidelines', they clearly do not. For example: references to product specifications, catalogue numbers and suppliers are still missing, the number of animals in the transport boxes from off-site breeding to the experimental unit, the handling method(s) used (per animal) for those animals included in experiment 2. 

Information on animal numbers, group sizes, distribution of sexes and age groups is not consistent. For example that kind of information appears to be missing for the behaviour analyses of animals that were bred in the off-site facility (212 animals). 

It is not clear from the data analysis paragraph and the results section whether the various uncontrolled variables (sex, age, strain and housing) have been sufficiently considered when doing the statistical analysis of the effects at the species level. Furthermore, it is not clear whether the individual comparisons are sufficiently powered. Therefore, the statistical significance of between group differences cannot be confirmed. 

Standard enrichment: enrichment is considered to be in addition to basic requirements. Bedding, diet, water and nesting material are considered basic requirements.   

The supplier of the antibody used for corticosterone concentration analysis is mentioned. However, a literature reference is missing. Information on specificity and sensitivity of the Ab and test should be given.   

Results: 

Supplementary table 1: the total animal number included is 212. In the table, the 'ease of handling' results of only 204 animals is reported. The discrepancy is not explained. 

Experiment 2: all animals were pooled for each method because 'individual group numbers were relatively small and each subgroup was fully counter-balanced'. This is not evident from the information provided in table 1. 

See comment under M&M about the statistical analysis and the differences between groups and whether or not they are statistically significant.

Discussion:

lines 444-447: explain 'we also saw changes in the measures taken before handling'? What are these measures and how do they suggest a reduced stress response? 

lines 447-450: explain how transport stress could have influenced the outcome of the hand approach test differently from the other tests?

In view of the comments made under M&M and Results about whether all variables were considered when using the statistical methods and the individual experiments were sufficiently powered, this reviewer is not convinced the conclusion 'at the population level, this study does suggest benefits to animal welfare' can be drawn.  

Author Response

Thankyou for the opportunity to respond to the reviewer’s comments on our manuscript. We also appreciate the time the reviewers have taken to provide helpful feedback and suggestions of how we can improve the presentation of the work. We have provided a response to each of the points raised below and also made changes to the manuscript which are highlighted in the revised version. We hope that these will address the reviewer’s concerns and the paper is now suitable for publication.

Editors comment

M&M

1)Although the authors state that they 'comply with ARRIVE guidelines', they clearly do not. For example: references to product specifications, catalogue numbers and suppliers are still missing, the number of animals in the transport boxes from off-site breeding to the experimental unit, the handling method(s) used (per animal) for those animals included in experiment 2. 

Apologies for what's missing. Number of animals in transport boxes has been added to line 129. Handling method of animals used in experiment 2 has been added to line 154.

The only potential place for a catalogue number and supplier that we can identify in the paper is for the CORT analysis but as stated in the paper, the materials for this were provided by a collaborator who is referenced in the paper and there is no catalogue number or supplier which can be included. Reference to previous studies using this antibody and demonstrating its validity and the group who provided it is included in the paper.

2) Information on animal numbers, group sizes, distribution of sexes and age groups is not consistent. For example that kind of information appears to be missing for the behaviour analyses of animals that were bred in the off-site facility (212 animals). 

In line 87 we have specified the number of animals used in each handling groups for experiment 1. We have also added total as well for clarity. In line 88 ‘As this experiment involved multiple research groups details of group size, sex and age were not recorded and therefore not considered in analysis’  We have also added age to line 93.

3) It is not clear from the data analysis paragraph and the results section whether the various uncontrolled variables (sex, age, strain and housing) have been sufficiently considered when doing the statistical analysis of the effects at the species level. Furthermore, it is not clear whether the individual comparisons are sufficiently powered. Therefore, the statistical significance of between group differences cannot be confirmed. 

As outlined in our introduction, the purpose of this study was to gain technicians perspective on the impacts of different handling method to support implementation of a change in policy to non-aversive handling methods. To make this feasible across a large number of animals and different technical staff whilst not adversely impacting on their daily workload, we kept the analysis as simple as possible and with a focus on our primary outcome – did handling methods in the breeding facility impact on technician handling-related scores during receipt at the experimental unit. We could also not balance the study design by sex, age or strain and so it would not have been appropriate to include these in the analysis. We feel this is clear in the paper and reflects our purpose and identifying sex differences for example was not our hypothesis. We have now added additional details on these limitations to the MS.

On line 471 ‘There are a number of limitations with this study design including the lack of data relating to sex, strain or age difference. However, Hurst et al has previously shown that non-aversive handling methods are beneficial on discrete measures. This experiment was designed to be a simple real world study to support the assessment at an institutional level of the benefits of different handling methods to implement policy change. At a population level, this study does suggest benefits to animal welfare.’

4) Standard enrichment: enrichment is considered to be in addition to basic requirements. Bedding, diet, water and nesting material are considered basic requirements.   

Line 102 standard enrichment was changed to standard housing conditions

5) The supplier of the antibody used for corticosterone concentration analysis is mentioned. However, a literature reference is missing. Information on specificity and sensitivity of the Ab and test should be given.   

The antibody has been widely used and reported in the literature and a reference to this previous work is provided (Ref 16). We do not have any further details which we can add.

Results: 

6) Supplementary table 1: the total animal number included is 212. In the table, the ‘ease of handling’ results of only 204 animals is reported. The discrepancy is not explained. 

This discrepancy has been corrected 212 animals were tested and analysed.

7) Experiment 2: all animals were pooled for each method because 'individual group numbers were relatively small and each subgroup was fully counter-balanced'. This is not evident from the information provided in table 1. 

See comment under M&M about the statistical analysis and the differences between groups and whether or not they are statistically significant.

Will need to see in the paper but maybe point above will clarify.

Hopefully answer to question 3 has clarified.

Discussion:

8) lines 444-447: explain ‘we also saw changes in the measures taken before handling’? What are these measures and how do they suggest a reduced stress response? 

Have clarified on line 448 that the measure taken before handling was ‘box opening’ which showed mice cup and tube handled mice were less responsive to box opening

9) lines 447-450: explain how transport stress could have influenced the outcome of the hand approach test differently from the other tests?

Hurst et al showed that using cupping or tube that mice were more likely to voluntarily approach the hand. We did not see this result and suggest that ‘transport stress immediately preceding the assessment may impact their baseline stress levels and mask more subtle behavioural outputs’

10) In view of the comments made under M&M and Results about whether all variables were considered when using the statistical methods and the individual experiments were sufficiently powered, this reviewer is not convinced the conclusion 'at the population level, this study does suggest benefits to animal welfare' can be drawn.  

We recognise that there are limitations to this work which we have discussed in the MS. This seems to have been acceptable to the other reviewer and we are not clear why this reviewer feels that drawing conclusions about animal welfare are not justified. Based on the measures we have taken, out data would suggest that non-aversive handling methods cause less overt signs of distress and are associated with animals being easier to handle by technical staff. Overt signs of distress remain one of the most widely used methods to assess procedure related endpoints and hence the welfare costs of interventions.

This manuscript is a resubmission of an earlier submission. The following is a list of the peer review reports and author responses from that submission.

Round 1

Reviewer 1 Report

The article by Davies et al entitled “Impact of refinements to handling and restrain methods in mice” is an interesting study showing that methods alternative to tail handling in the context of regular husbandry procedures have a positive impact on the acute behavioral response to handling as well as anxiety-related behaviors and circulating corticosterone in mice. While the manuscript is generally well written and the study design is good, I have some comments/questions:

Introduction

  1. In lines 66-67, it is stated that “to investigate the impact of handling methods on the experience of animal care staff…”. This makes it sound as the study was an assessment of the perception of animal care staff regarding different handling methods (i.e. how would they be willing to adopt these new procedures) , but that’s not really the case, as the study focuses on the effects these handling methods have on the mice.
  2. On that same note, it is a bit confusing as to what the real novelty/objective of this study is. Is this the first study assessing the behavior of the animals in response to handling associated with husbandry procedures?

Materials and methods

  1. Since there are several different experiments, it would be useful to have a timeline for each of them. Right now, is a bit confusing as to which animals were used for what experiment and when.
  2. Where the animals group or single housed?
  3. Where the different handling methods performed in different facilities, or the three of them were used in each of the offsite facilities mentioned?
  4. Were all the animals the same age when tested?
  5. How were the animals from different facilities transported back? (i.e. vehicle type, distance, time of the day)
  6. If the three of them were used in each of the facilities, were each the different handling methods associated with specific rooms and/or staff, as well as frequency and/or time of handling?
  7. In lines 99-101, what do you mean with “offspring only”? And what do you mean with “defined by holding room”?
  8. A diagram showing each of the handling methods- tail, cupping, and tube- would be really helpful
  9. On that same note, were staff trained to perform each of the handling methods optimally and in a standardized way?
  10. Why were all mice handled using the cup method upon reception? This could bias the results as one of the experimental group uses the same method (i.e. the animals could show reduced aversion as a result of being used to the method, not because they are overall less anxious).
  11. Was the receiving technician the same person for all of the mice?
  12. The hand approach test is not described in the methods
  13. The name “non-tail” restrain method is a bit vague. It would be useful for future references to have a name that better describes this novel and useful technique
  14. Lines 125-127 are the same as 120-122
  15. The conditioned place preference methods and goal are confusing.

Results.

  1. Can you add individual datapoints to the graphs in figure 2?
  2. Did you see any sex difference?
  3. Figure 3 is missing the legends. What does the gray vs white bars mean?

Discussion

Since the significant differences between handling methods were mostly seen only in the acute behavioral response to handling, the conclusions of the study that “non-tail handling methods can benefit mouse welfare” are a bit too ambitious., as not measures of in cage behavior and overall health were provided.

Reviewer 2 Report

The authors report the results and conclusions of a study in which they score the behaviours of two strains of mice reared off site upon arrival at the experimenters site. They also report on the effects of a refined restraining method including performance in a behavioural test. Plasma corticosterone was determined in two cohorts of mice exposed to tail versus non-tail restraint and the mice used in the behavioural test. 

The manuscript is well written. The study design and analysis methods are appropriate for answering the research questions. However, I have a few questions and some suggestions to make. 

Materials and Methods

  1. Animals and Housing:
    1. List for all cohorts per strain the number of males and females. Already some of this information is mentioned in the M&M, but for other cohorts one has to go to figure legends. 
    2. List the age of the animals especially in relation to the cohort of 'mature' CD1 males. Mature refers to which age range? 
    3. Where applicable list housing density.
    4. Describe transport: duration, means, etc. Important since transport stress is mentioned in the discussion (line 345).
    5. Line 83: C57Bl/6 typo should be capital L.
    6. Line 87: introduce abbreviation 'CORT' 
    7. Line 93: List Relative humidity of the rooms.
  2. Evaluation of different handling methods
    1. Line 100-101: defining the three different handling groups by holding room could have introduced a confounding factor (room). Have you checked for the presence of a confounding effect? 
    2. Line 107-108: what was the rational for choosing the cupping method for all animals upon arrival? 
  3. Non-tail restraint method ...
    1. Line 128: table 2: make explicit that the vocalisations are those in the audible range (for humans) unless ultrasound vocalisations were recorded as well. 
  4. Enzyme immunoassay of serum corticosterone
    1. Line 160: what is the antibody? Provide a literature reference?

Results

  1. Effect of off-site breeding facility handling methods ...
    1. Results are presented comparing handling methods ignoring strain and sex differences. Provide rational in results and/ or discussion.
    2. Line 221: lower scores while they are actually higher scores representing the 'calm/ minimal' response. 
    3. Lines 223 and 226: correct the references made to the panels: B should be C and C should be E. 
  2. Modified restraint method reduces overt ...
    1. Line 238: the significant difference between restraint method for mature CD1 males is mentioned. However, a similar significant difference is present in B6 males according to fig. 3. 
    2. Line 251 and 252: lower scores for struggling and lower occurrences of vocalisation - shouldn't that be 'higher' in both. 
    3. Line 256: the commonly used abbreviation for C57BL/6 is B6.
    4. Fig. 3 gives the impression that some of the B6 male and female responses differ from those of male and female CD1 mice. 
    5. Fig. 3  panel F: the order of white and grey bars is different from the other panels. 
  3. Tail versus non-tail restraint ...
    1. Fig. 4 panel A: no indication of which two bars represent tail and which two represent non-tail restraint. 
    2. Fig. 4 panel B and D: Y-axes: there is a $ sign in brackets which should be a %. X-axes: replace 'arm' with 'non-tail'.
  4. Different restraint methods ...
    1. Line 301 and 302: C57BL/6J
    2. Fig. 5 panel B: Y-axis: add to 'frequency' 'cotton bud biting'
    3. Is the cotton bud biting test an appropriate test given the significant spreading of the scores of the individual animals? 
    4. Why were only males tested for the CPP test? 

Discussion

  1. Line 343-346: what was potentially different about the way the test was carried out? Elaborate on the transport stress as a causing factor (in relation to the more detailed information on the transport logistics and means to be provided under M&M). 
  2. Line 349: 'what' shouldn't that be 'that'?
  3. Line 357: 'in' shouldn't that be 'is'?
  4. Line 376: Could it be that the stress caused by the EZM masks the effects of the different handling methods? 
  5. Lines 386-388: the explanation given for the lower biting frequency shown by the CD1 mice seems counter intuitive? 

Conclusions

  1. This section does not appear to include all conclusions that can be drawn from the experimental data. 

References

  1. References 10 and 14 appear to be identical. 

Reviewer 3 Report

This is an interesting and well written paper.  It shows several interesting results.  Two non-tail handling catching methods were compared with tail handling.  There were changes in the behaviour indicating that the animals were less stressed by the non-tail handling methods, but the corticosterone levels did not reflect any changes.

I have the following specific comments.

Line 10, 315 and elsewhere: Stressors cause the emotional state of distress to an animal.  Some stressors can have adaptive effects and positive effects - eustress. Often the word stress in this paper is used when they mean distress. Would the authors please check.

Line 175: what was used to rinse the equipment  

Line 128: why do they not measure ultrasound as that is a common frequency for alarm signals  

They use tunnel and tube, better to stick to one term or explain interchangeability in the M&Ms  

They need to add where they bought the EZM and CPP from

Line 337: I note that the authors also wanted to look at any stress response  i.e. hormonal difference.

Lines 353 et seq.  It would be helpful to readers to consider that this in turn may benefit the science as the mice were easier to handle with severer procedures such as tail bleeds, gavage, and will less distressed with the consequent hormonal disturbances.

Line 372: et al. not et al

Do the authors have any control corticosterone data for levels in control non-transported mice?

The authors might like to consider the following older papers in regard to the subject of their paper. 

TULI, J . et al., (1995) Corticosterone, adrenal and spleen weight in mice after tail bleeding, and its effect on nearby animals.  Laboratory Animals 29, 90-95.

TULI, J, et al., (1995) Stress measurements in mice after transportation. Laboratory Animals 29, 132-138

TULI, et al., (1995) Effects of acute and chronic restraint on the adrenal gland weight and serum corticosterone concentration of mice and their faecal output of oocysts after infection with Eimeria apionoides. Research in Veterinary Science 59: 82-86

The reasons I am suggesting the authors to look at the papers by Tuli et al is that they also found behavioural changes without any detectable change in corticosterone levels.  This is important as it places a different emphasis on the evidence for any adverse effects on behaviour rather than stress hormones.  Important for the US market?

Round 2

Reviewer 1 Report

I still think the questions that this study is trying to answer are highly timely and relevant. Nonetheless, while the authors addressed many of the minor suggestions, now that the methods are clearer I still have major concerns that, in my opinion, make this study unpublishable unless additional studies are run (to replicate data), or major edits in the interpretation of the results are made. Please see below.

  1. If the different handling methods were performed in different rooms, it is unfortunately impossible to know whether the effects seen are due to the handling methods or the room were animals are housed. While I understand that the rooms can be identical from a human perspective, there could be imperceptible variables, like ultrasonic noises, that likely differ between the rooms. As another possible variable influencing the behavior of the mice is the order in which rooms are visited by the staff, for example (see Lauer et al, 2009). If you wanted to keep the treatments limited to specific rooms to make it easier for the staff, you should have run multiple cohorts in which each treatment/room was randomized. Alternatively, you should have randomly assigned cages to each treatment (having all treatments in each room) and include the individual cages as a variable in a nested ANOVA (at least in animals that were group housed). If you want to publish these data, I strongly suggest adding a new cohort in which each treatment has been assigned to a different room.
  2. I also understand that you want to see the effects of your manipulations at the species level, but if you want to do so, you first need to show that there are no sex or strain differences in the dependent variables you are measuring. Studies have shown that handling and enrichment protocols can have different effects depending on sex and strain (see Sensini et al, 2020, Minie et al, 2021, and Nevison et al, 1999). You should at least consider these as covariates using alternative statistical methods, especially considering the high variability you have in your data.
  3. Finally, stating that your study “demonstrates that non-tail handling methods can benefit mouse welfare” is a serious overinterpretation of your results, even if your study design and statistical methods had been appropriate. This, because you only show significant differences in the behavioral response to handling, but no other behavioral or physiological measure.